# Oleanolic Acid Complexation with Cyclodextrins Improves Its Cell Bio-Availability and Biological Activities for Cell Migration

**DOI:** 10.3390/ijms241914860

**Published:** 2023-10-03

**Authors:** Javier Stelling-Férez, Santiago López-Miranda, José Antonio Gabaldón, Francisco José Nicolás

**Affiliations:** 1Department of Nutrition and Food Technology, Health Sciences PhD Program, Universidad Católica de San Antonio Murcia (UCAM), Campus de los Jerónimos n°135, Guadalupe, 30107 Murcia, Spain; javier.stelling.ferez@gmail.com (J.S.-F.); slmiranda@ucam.edu (S.L.-M.); jagabaldon@ucam.edu (J.A.G.); 2Regeneration, Molecular Oncology and TGF-β, Instituto Murciano de Investigación Biosanitaria (IMIB)-Arrixaca, Hospital Clínico Universitario Virgen de la Arrixaca, El Palmar, 30120 Murcia, Spain

**Keywords:** hydroxypropyl cyclodextrins (HP-CDs), oleanolic acid (OA), cell migration, MAP kinases, epidermal growth factor receptor (EGR), c-Jun, wound healing

## Abstract

Wound healing is a complex process to restore skin. Plant-derived bioactive compounds might be a source of substances for the treatment of wounds stalled in a non-resolving stage of wound healing. Oleanolic acid (OA), a pentacyclic triterpene, has shown favorable wound healing properties both in vitro and in vivo. Unfortunately, OA cannot be solubilized in aqueous media, and it needs to be helped by the use of dimethyl sulfoxide (DMSO). In this paper, we have shown that cyclodextrins (CDs) are a good alternative to DMSO as agents to deliver OA to cells, providing better features than DMSO. Cyclodextrins are natural macromolecules that show a unique tridimensional structure that can encapsulate a wide variety of hydrophobic compounds. We have studied the cyclodextrin-encapsulated form of OA with OA/DMSO, comparing their stability, biological properties for cell migration, and cell viability. In addition, detailed parameters related to cell migration and cytoskeletal reorganization have been measured and compared. Our results show that OA-encapsulateds compound exhibit several advantages when compared to non-encapsulated OA in terms of chemical stability, migration enhancement, and preservation of cell viability.

## 1. Introduction

The skin acts as a physical barrier against pathogens, ultraviolet radiation, or temperature [1], and its integrity is restored through the coordinated action of different signaling molecules and cell types in a process called wound healing [2,3,4]. Complications in wound healing may cause an uncomplete wound closure that may lead to an ulcer [5,6], a lesion that severely complicates a patient’s life quality [7,8,9,10,11], and it is a great burden for health care systems for its high costs and time-consuming treatments [12,13,14].

Plant-derived bioactive compounds arise as a good alternative for the treatment of complex wounds because of their potential to improve cell migration [15,16,17]. Among them, oleanolic acid (OA) is a pentacyclic triterpene that has shown promising wound healing properties in vitro and in vivo [18,19]. Indeed, it has been shown that OA enhances cell migration, a crucial process for wound healing. Concretely, the test of OA in various in vitro scratch assays with cell migration models such as Mink lung epithelial (Mv1Lu) and Human Mammary Gland (MDA-MB-231) cell lines has demonstrated its important pro-migratory effect, which is comprised by a complex kinase regulatory system that turns on in response to OA [20,21]. Thus, OA induces the phosphorylation of epidermal growth factor receptor (EGFR), triggering the activation of both extracellular signal-regulated kinase (ERK1/2) and c-Jun N-terminal kinase (JNK1/2) [20,21,22,23]. As a consequence, OA stimulation leads to the activation of a master regulator for cell migration, c-Jun, a transcription factor [24,25,26,27]. Specifically, in response to OA, c-Jun is mainly phosphorylated and overexpressed at the scratch edge of in vitro scratch assays both in Mv1Lu and in MDA-MB-231 cells [21]. Together with inhibitor experiments, this indicates that c-Jun participates in OA-enhanced cell migration. Additionally, these pathways are intimately related to other proteins in charge of cytoskeleton remodeling: focal adhesion kinase (FAK) and paxillin [28,29], which interact to execute an assembling-disassembling loop of actin cytoskeleton and focal adhesions (FAs), structures that allow migration of epithelial cells [30,31,32,33]. Thus, we have shown that OA promotes paxillin remodeling, increasing the number of FAs and decreasing FA size, a result reinforced by the fact that OA also promotes FAK-paxillin interaction [21]. These results indicate a high cytoskeleton dynamization induced by OA, needed for cell migration [34]. Because of these, in vitro scratch assays with Mv1Lu and MDA-MB-231 can be used as good biosensors for the improvement of OA application in migration as a foreseeable use in human skin wound healing [20,21,23,35,36].

The main constraint for OA application is its lipophilic nature. Usually, OA can be solubilized in aqueous solutions by using dimethyl sulfoxide (DMSO). Unfortunately, DMSO has mild cytotoxic effects, which compromise cell viability [37,38]. In addition to this, meticulous care is needed to manage OA/DMSO solutions in cell assays due to the fact that OA often precipitates and crystalizes, loosing bioavailability [20,21]. To help with all these issues, OA can be complexed with cyclodextrins, which allows it to be carried into aqueous solutions [39]. Cyclodextrins are natural macromolecules that show a unique tridimensional structure; they have a hydrophilic surface that confers high water solubility and a hydrophobic cavity that encapsulates a wide variety of hydrophobic compounds [39,40,41,42]. Many authors have shown the potency of cyclodextrins to deliver drugs since they reinforce their biological activities [39,43,44,45,46,47]. There are three types of native cyclodextrins, α, β, and γ, that differ in the number of D-glucose units that formed them, 6, 7, or 8, respectively, which progressively increase the size of the hydrophobic cavity [48]. Additionally, chemical groups can be added to native cyclodextrins for several purposes, such as increasing their water solubility [37,49,50,51]. Interestingly, using native cyclodextrins (α-, β-, and γ-CDs) and the same ones with hydroxypropyl groups (HP-α-, HP-β-, and HP-γ-CDs), a recent study determined which cyclodextrin had the highest performance to encapsulate OA [52]. Complexation constant (K_c_) and encapsulation efficiency (EE) have been used to evaluate OA complexation, which revealed the highest K_c_ for β- and γ-CDs, followed by their homologues HP-β- and HP-γ-CDs [52]. Seeking a compromise between adequate complex stability and sufficiently high solubility of oleanolic acid, the modified cyclodextrins HP-β- and HP-γ-CDs emerged as the most suitable candidates for complex formation. However, the biological activity of the OA/HP-β-CD and OA/HP-γ-CD complexes has not been tested in vitro yet. In this paper, we have shown that OA can be maintained stable in a conjugated and dehydratable form to be applied to biological hydrophilic systems. Using cell and biochemistry models, we have assessed the effect of the complex on migration and cell viability, finding a better performance than the DMSO solubilized form. Therefore, a future application of these complexes to wound healing is more feasible.

## 2. Materials and Methods

### 2.1. Preparation of Oleanolic Acid in DMSO

Oleanolic acid (OA, purity >97%) (Merck, Darmstadt, Germany) was dissolved in dimethyl sulfoxide (DMSO) (Merck, Darmstadt, Germany) to a 25 mM concentration. Final assay concentrations are indicated for each experiment in the figure legends. The DMSO concentration in all assays never exceeded 1% to avoid cytotoxic effects. In any case, a DMSO control condition was performed in each experiment.

### 2.2. Oleanolic/Cyclodextrin Complexes Preparation: Solid Complexes by Freeze Drying

The modified hydroxypropyl beta and gamma (HP-β- and HP-γ-) cyclodextrins (CDs) were used to encapsulate OA (Winplus International Limited, Ningbo, China). OA and CDs were added to 250 mL of aqueous solutions with a molar ratio of 0.2/50 (OA/CD) for both HP-β- and HP-γ-CDs. These molar proportions were selected according to phase solubility diagrams previously published by López-Miranda et al. (2018) [52]. Phase diagrams allow for the determination of the optimal molar ratio between cyclodextrin and the guest molecule to maximize their dissolution. Furthermore, the stoichiometry of the complexes formed is 1:1, indicating that when the complex is formed, one molecule of cyclodextrin complexes with a single molecule of oleanolic acid [52]. Solutions were constantly stirred for 24 h at 20 °C in the dark for complex formation in aqueous solutions. After that, OA/CD solid complexes were obtained by dehydrating a 250 mL volume of dissolved complexes by freeze drying. Solutions of OA/CD complexes (OA/HP-β-CD and OA/HP-γ-CD) were dehydrated in a Christ Alpha 1–2 LD Plus (Martin Christ, Osterode am Harz, Germany) freeze dryer at −48 °C for three days. This step was critical to obtaining an optimal formulation for its application to in vitro cell culture (see Appendix A). The recovered freeze-dried solid complexes were stored in plastic containers protected from light for posterior analysis and characterization.

The dehydration yield (DY) was calculated using the following equation:$$DY\left(\%\right)=\frac{solid\;complexes\;obtained\left(g\right)}{total\;solids\;in\;solution\left(g\right)}\cdot 100$$

The encapsulation efficiency (EE) was calculated using the equation:$$EE\left(\%\right)=\frac{total\;compound\;encapsulated\;in\;solid\;complex\left(mg\right)}{initial\;total\;compound\;in\;solution\left(mg\right)}\cdot 100$$

The drug load (DL) was calculated using the equation:$$DL\left(mg\;g^{-1}\right)=\frac{total\;compound\;encapsulated\;in\;solid\;complex\left(mg\right)}{total\;solid\;complexes\left(g\right)}$$

Final assay OA/CD complex concentrations are indicated at each experiment in figure legends, which indicate their molar ratio in terms of µM for OA and mM for both HP-β-CD and HP-γ-CD (µM/mM).

### 2.3. Oleanolic Acid Determination by HPLC

OA that was complexed in cyclodextrins was quantified by High-performance liquid chromatography (HPLC) analysis, using an HPLC Agilent Technologies model 1200 (Santa Clara, CA, USA). Firstly, a calibration line was set with 1 mg/mL free OA (OA, purity >97%) (Merck, Darmstadt, Germany) to refine the quantification. Then, complexed OA was analyzed by HPLC equipped with a DAD detector set at a 210 nm wavelength, injecting 20 µL of complexed OA. Separations were performed on a plugged (5 µm) HPLC Catridge 250-4 LiChospher 100 RP-18 (Sigma-Aldrich, St Louis, MO, USA). The column temperature was set to 30 °C, and the flow rate was 1 mL/min. The mobile phase used was 20% water and 80% acetonitrile for a total running time of 30 min. During this time, an OA-specific absorbance peak at 210 nm appeared for close to 10 min. Finally, the data were processed, and OA concentrations were expressed in OA mg per CD g; this is the drug load (DL), as defined above.

### 2.4. Cell Culture

Mink Lung Epithelial (Mv1Lu) [36,53,54] cells were grown in Eagle’s Minimum Essential Medium (EMEM) (Biowest, Nuaillé, France) supplemented with 10% Fetal Bovine Serum (FBS, Gibco, Thermo Fisher Scientific, Rockford, IL, USA), 1% Penicillin/Streptomycin, and 1% L-Glutamine (both from Biowest, Nuaillé, France). Mv1Lu cells were cultured in an incubator at 37 °C in a 5% CO_2_-humified atmosphere. Human mammary gland cells (MDA-MB-231) [23,35] were grown in Dulbecco’s Modified Eagle Medium (DMEM) supplemented as mentioned above for EMEM. MDA-MB-231 cells were incubated in an incubator at 37 °C with a 7.5% CO_2_ humified atmosphere.

### 2.5. Cell Proliferation Assay

Cell proliferation over 10 days was evaluated by cell counting. Essentially, cells were seeded in 5 cm diameter plates at a density of 10^5^ cells/plate in a complete medium supplemented with 10% FBS. Mv1Lu or MDA-MB-231 cells were treated in duplicate with DMSO, OA/DMSO, HP-β-CD, and OA/HP-β-CD; final concentrations are indicated in figure legends. At even days until day 10, cells from all conditions were detached using trypsin/EDTA, and the viable cell count was calculated using a TC10 automated cell counter (BioRad, Hercules, CA, USA) and a trypan blue dye (Sigma-Aldrich, St Louis, MO, USA) exclusion assay. Data were gathered and plotted in Graph Pad Prism 7 software as a growth curve.

### 2.6. Cell-Front Migration Assay, Subcellular Localization Assay

Mv1Lu cells were grown until they reached confluence on round glass coverslips in 10% FBS-EMEM medium. At this time, cells were washed, and a 24 h serum-starvation period was set by replacing the medium with FBS-free EMEM. After this, the epithelium was scratched using a razor blade, which produced an artificial wound (it will be called “scratch” throughout the paper) with enough space to allow cells to migrate. The newly scratched wound was set as time 0 h of the experiment, and then 5 µM OA/DMSO or 12.5/3.125 µM/mM OA/HP-β-CD were added to the medium. As parallel conditions, DMSO or HP-β-CD (3.125 mM) equivalent volumes were added as vehicle controls. After the selected times of incubation, glass coverslips were fixed with 4% formaldehyde (Applichem GmbH, Darmstadt, Germany) in PBS (Biowest, Nuaillé, France) for 10 min and washed twice with PBS. After this, cells were permeabilized with 0.3% Triton X-100 (Sigma-Aldrich, St. Louis, MO, USA) in PBS for 10 min. For immunostaining, blocking was performed for 30 min at room temperature in a PBS solution with 10% FBS, 5% skim milk (Beckton Dickinson, Franklin Lakes, NJ, USA), 0.3% bovine serum albumin (BSA, Sigma-Aldrich, St Louis, MO, USA), and 0.1% Triton X-100. Then, cells were incubated for 1 h at room temperature with anti-paxillin or anti-phospho-c-Jun antibodies, diluted in blocking solution without skim milk. Proper fluorescently labeled secondary antibodies (see Section 2.9) were co-incubated for 30 min with Alexa Fluor 594 conjugated phalloidin (Molecular Probes, Thermo Fisher Scientific, Waltham, MA, USA) and Hoechst 33258 (Fluka, Biochemika, Sigma-Aldrich, St Louis, MO, USA) to reveal actin cytoskeleton and nuclei, respectively. Once samples were inmunostained, image acquisition was performed with a confocal microscope (LSM 510 META from ZEISS, Jena, Germany). The setting of images was performed using Zeiss Efficient Navigation (ZEN) interface software (ZEISS, Jena, Germany). When a wider view of the migration front was required, concretely in phospho-c-Jun staining (indicated in Figures), 4 × 3 linked fields were acquired by the “Tile scan” ZEN tool. Subsequently, tile scan fluorescent signals were converted to a linear mode and covered with a full data range using the Rainbow look up table (Rainbow LUT) in Image J software. In order to quantify phospho-c-Jun levels in immunofluorescence pictures, images were analyzed and quantified by Image J software. For this purpose, pictures in 8-bit, three-channel format (Red, Green, Blue, and RGB) were divided into three separate color channels (three pictures). By using the blue channel picture (Hoechst staining), regions of interest (ROIs) were established to define each nucleus, creating as many ROIs as nuclei in the image. Then, by overlapping these masks onto the corresponding green channel picture (p-c-Jun staining), we calculated the green intensity value of each nucleus (ROI). Because of the large area covered in each picture (tile scan), they were divided into three equal sectors (S1, S2, and S3), with S1 being the outermost edge on the scratch (Appendix A). Within each sector, the quantified signal of each nucleus was used as a replicate to obtain p-c-Jun intensity data in each of the conditions performed. For a better understanding, the relation between the number of p-c-Jun positive nuclei and the total nuclei number was calculated. For this purpose, a basal p-c-Jun intensity mean was calculated considering the control condition of the assays, set as a threshold. In this sense, nuclei with p-c-Jun intensity over the threshold were set as positive nuclei. On the other hand, the “Z stack” ZEN feature was used when deep cytoskeleton structure observance was required, taking a proper number of pictures along the Z axis. Focal adhesion (FA) quantifications were performed as described in Horzum et al., 2014 by using CLAHE and Log3D macros for ImageJ [55]. Briefly, focal adhesions (FAs) were quantified from paxillin-stained acquired pictures. We used three different images for each condition. Specifically, cell filopodia were selected as regions of interest (ROIs), and the resulting areas (containing FAs) were considered for further analysis. A number of five filopodia were considered for each picture. Then, the number of FAs was calculated in each filopodia by using the previously mentioned macros. The obtained number was divided by the total filopodia area to determine FA density. In parallel, the size of each FA was measured using the macros mentioned above. Finally, the FA average size was calculated for each treatment.

### 2.7. In Vitro Scratch Assay

Mv1Lu or MDA-MB-231 were seeded in 24-well plates until they reached 100% confluence, using the appropriate medium for each line with all supplements. At that time, the medium was changed to a FBS-free medium for 24 h. At the initial time (time 0 h), a cross-shaped scratch was made on the cell monolayer using a sterile p-200 µL pipette tip. After replacing FBS-free culture medium to wash out released cells, OA, OA/HP-β-CD, or OA/HP-γ-CD (indicated concentration at each figure legend) were added. An equivalent volume of DMSO, HP-β-CD, and HP-γ-CD was added. In parallel, a positive control was set by adding 10 ng/mL epidermal growth factor (EGF, Sigma-Aldrich, St. Louis, MO, USA). Additionally, pharmacological inhibitors against key proteins on migration regulation were added to OA conditions: 2.5 µM PD153035 (Epidermal Growth Factor inhibitor, EGFRi), 50 µM PD098059 (Mitogen-activated protein kinase kinase inhibitor, MEKi), and 15 µM SP600125 (c-Jun N-terminal kinase inhibitor, JNKi) [56,57,58]. After a 24-hour incubation period, cells were fixed with 4% formaldehyde (Applichem GmbH, Darmstadt, Germany) in PBS (Biowest, Nuaillé, France) for 10 min. Finally, well plates were washed twice with PBS. Pictures were taken at 10× magnification using an optical microscope equipped with a digital camera (Motic Optic AE31, Motic Spain, Barcelona, Spain). To quantify cell migration, the areas of the gaps in the scratches were measured by ImageJ software. The initial cell-free surface (time 0 h) was subtracted from the endpoint cell-free surface (time 24 h) and plotted in a graph [59].

### 2.8. Western Blot

Mv1Lu or MDA-MB-231 cells were seeded and allowed to reach 60% confluence in 10 cm diameter plates. At this time, culture medium containing 10% FBS was replaced by an FBS-free medium, incubating the cells for a 24-hour period. Serum-deprived cells were treated with either OA, OA/HP-β-CD, DMSO, HP-β-CD, or 10 ng/mL EGF. After time incubations, cells were harvested, washed twice with ice-cold PBS, and lysed with 20 mM TRIS pH 7.5, 150 mM NaCl, 1 mM EDTA, 1.2 mM MgCl2, 0.5% Nonidet p-40, 1 mM DTT, 25 mM NaF, and 25 mM β-glycerophosphate supplemented with phosphatase inhibitors (I and II) and protease inhibitors (all from Sigma-Aldrich, St Louis, MO, USA). The total protein amount of all extracts was measured and normalized by the Bradford assay [60] (Sigma-Aldrich, St. Louis, MO, USA). The extracts were analyzed by SDS-PAGE followed by western blot (WB) using the indicated antibodies. Blots were revealed by using Horseradish peroxidase substrate (ECL) (GE Healthcare, GE, Little Chalfont, UK), and images were taken with a Chemidoc XRS1 (Bio-Rad, Hercules, CA, USA). For protein quantification, western blot pictures in 8-bit format were processed in ImageJ software. Subsequently, in all western blot pictures, a lane was established for each of the samples. In each lane, the band was selected according to the specific size (kDa) of the protein of interest. For each total protein and their phosphorylated version, each band’s intensity peak was plotted, and next, the area under the plot was measured by using the “Wand (tracing) tool” of ImageJ to obtain the intensity value. In order to normalize, obtained intensity values were referred to as obtained intensity values of either the unphosphorylated form of the protein (total) or a loading control protein such as β-actin, only if the unphosphorylated form was undetectable (non-available antibody for detecting the unphosphorylated form).

### 2.9. Antibodies

The following commercial primary antibodies were used: anti-phospho-ERK1/2, anti ERK1/2, anti-JNK1/2, anti-phospho-JNK1/2 and anti-phospho-c-Jun (all from Cell Signaling Technology, Danvers, MA, USA); anti-phospho-EGFR (Thermo Fisher Scientific, Rockford, IL, USA); anti-EGFR, anti-paxillin and anti-c-Jun (Santa Cruz Biotechnology, Heidelberg, Germany); and anti-β-actin (Sigma-Aldrich, St Louis, MO, USA). Secondary antibodies were anti-rabbit IgG Horseradish peroxidase-linked F(ab’)2 I fragment (from donkey) (GE Healthcare, GE, Little Chalfont, United Kingdom), anti-mouse IgG_1_ (BD Pharmingen, Beckton Dickinson, Franklin Lakes, NJ, USA), and Alexa Fluor 488 conjugated anti-mouse (from donkey) (Thermo Fisher Scientific, Rockford, IL, USA).

### 2.10. Statistical Analysis

All the collected data were analyzed using Graph Pad Prism 7 software. In every analysis, classic statistical parameters were calculated and statistical tests were performed with a 95% confidence interval, consequently, *p*-values lower than 0.05 were considered to be statistically significant. In the figure legends, the asterisks denote statistically significant differences between conditions (* *p* < 0.05, ** *p* < 0.005, *** *p* < 0.001, and **** *p* < 0.0001). Data on intensity values, collected from western blots, were analyzed by a one-way ANOVA test, comparing the mean of each condition with the mean of every other condition. Then, a Tukey’s multiple comparisons test was performed. The same parameters and tests were applied for migration percentage values of scratch assays.

Data of intensity values obtained from p-c-Jun nuclei quantifications in immunofluorescence pictures were analyzed by using two statistical tests: a two-way ANOVA test and a one-way ANOVA test. Concretely, a two-way ANOVA following Tukey’s multiple comparison test was performed to compare the p-c-Jun intensity mean between sectors from different conditions (e.g., S1 DMSO versus S1 OA). On the other hand, a One-way ANOVA, following Tukey’s multiple comparison test, was performed to compare the p-c-Jun intensity mean between sectors from the same condition (e. g., S1 DMSO versus S2 DMSO).

## 3. Results

### 3.1. HP-β- and HP-γ- Cyclodextrins SHOW High Rates Complexation Parameters with OA

Different parameters of the dehydrated by freeze drying solid complexes of OA with HP-β- and HP-γ-CD were measured (Table 1).

The dehydration yield (DY) values were around 90% for both CDs. These values indicate a high level of complex recovery and were similar to those previously obtained in complex dehydration using the freeze drying technique [43].

Encapsulation Efficiency (EE) represents the percentage of OA that has been recovered after the dehydration process, indicating whether significant compound losses occur during the dehydration process. On average, EE was 80.0% for HP-β-CD, and 91.9% for HP-γ-CD. The amount of OA finally encapsulated per gram of complex (DL mg/g) was 1.06 and 1.17 for HP-β- and HP-γ-CD complexes, respectively (Table 1).

These results show that properly encapsulated solid complexes of oleanolic acid with HP-β- and HP-γ-CDs can be obtained, allowing for subsequent dosing and handling in an aqueous medium.

### 3.2. Freeze-Dried OA/HP-β-CD and OA/HP-γ-CD Complexes Stimulate Migration in Mv1Lu Cells

To measure OA/HP-β-CD and OA/HP-γ-CD complexes migration activity, we used mink lung epithelial cells, Mv1Lu, a very well-known epithelial model for the study of cell motility and cytoskeleton structures. They are also known for stopping proliferation when the cells reach confluence [36,54,61]. In previous papers, we have shown that OA 5 µM in DMSO stimulates Mv1Lu motility [20]. However, OA/DMSO levels beyond this concentration cause cell viability loss and cytotoxic effects [20,21,62]. It should be noted that the OA complexes used in the following in vitro assays were dehydrated by freeze drying to obtain a lyophilized powder compatible with in vitro cell culture. In fact, non-dried OA complexes did not exert any activity on cell migration (Appendix A). However, freeze-dried OA/HP-β-CD and OA/HP-γ-CD complexes clearly stimulated Mv1Lu migration from the scratch edges (Figure 1a).

Consequently, migration percentages with OA/HP-β-CD and OA/HP-γ-CD complexes were statistically significant compared to basal conditions (C) (Figure 1b) and also to vehicle controls (empty HP-β-CD and HP-γ-CD) (Figure 1a,b). In particular, these complexes showed the highest migration activity at 12.5/3.12 µM/mM and 125/31.25 µM/mM for OA/HP-β-CD and OA/HP-γ-CD, respectively (Appendix A). Furthermore, no statistical differences were found between OA/DMSO and OA/CDs on Mv1Lu migration. However, a slight tendency was observed for OA complexes suggesting a higher migration activity than OA/DMSO, concretely for OA/HP-γ-CD complexes at 125/31.25 µM/mM concentration (Appendix A). Interestingly, when looking at a wider view at the scratch assay, we noticed a clear larger recruitment of Mv1Lu migrating cells in the OA/CDs condition than in the OA/DMSO condition, suggesting a more extensive and improved migratory effect of complexed OA with cyclodextrins (Appendix A).

All these data suggest a powerful effect of OA complexed with HP-β- and HP-γ-CDs on Mv1Lu cell migration. However, among the two cyclodextrins studied, it was deduced that HP-β-CD allows a better release of OA from the complex because a lower OA concentration (12.5 μM vs. 62.5 μM) is required for a similar cellular migration when HP-β-CD is used for complexation instead of HP-γ-CD. So then, the following experiments were performed exclusively with the OA HP-β-CD complexed form.

### 3.3. Complexed OA at Freeze-Dried OA/HP-β-CDs Is Stable at Room Temperature

Conservation of OA in DMSO has always been a difficulty when the use of oleanolic in further applications has been envisaged. Once oleanolic acid is dissolved in DMSO, it needs to be preserved in liquid nitrogen. This is because factors such as exposure to light, oxygen, and temperature can influence its stability and reactivity. Generally speaking, molecule complexation with cyclodextrins adds an extra level of stability to such molecules [63,64]. Thus, after producing the OA HP-β-CD complexes, we tested OA stability in the complex that had been stored at different temperatures for different durations (Table 2). 

Essentially, evaluation by HPLC of complexes showed that, after 8 weeks of conservation at different temperatures (−80 °C, 4 °C, or 20 °C), cyclodextrin-loaded OA values were very close to the initial ones, even in the less favorable case of conservation at room temperature (20 °C) (Table 2). All these data suggest that after HP-β-CD complexation, OA becomes a very stable molecule.

### 3.4. Freeze-Dried OA/HP-β-CD Complexes Do Not Compromise Cell Viability

According to previous reports, OA/DMSO has a mild antiproliferative effect on epithelial cells [20,65,66], so we decided to check this outcome when OA was complexed with cyclodextrins. We performed a cell proliferation assay by culturing Mv1Lu in supplemented medium with 5 µM OA/DMSO and 12.5/3.12 µM/mM OA/HP-β-CD for 10 days and counting total cells each 2 days. Generally speaking, although a slight decrease in proliferation was noticed with OA/HP-β-CD by 6 and 8 days, at longer times, cyclodextrin vehiculation produced higher cell counts than the OA/DMSO condition (Figure 1c), being less counterproductive for cell proliferation when compared to OA/DMSO.

All these data suggest a less antiproliferative effect of OA complexed with HP-β- and HP-γ-CDs on Mv1Lu cell migration that exhibits an improvement in cell migration and better cell viability than the ones observed for OA/DMSO.

### 3.5. Specific Inhibitors against Migration-Related Proteins Decreased OA Migration Triggered by OA/HP-β-CD Complexes

Previously, it has been shown that OA/DMSO produces migration on Mv1Lu cells by activating the EGFR receptor and mitogen-activated protein kinases (MAP kinases) (mitogen-activated protein kinase kinase, MEK, and c-Jun N-terminal kinase, JNK) [18,20,21]. The role of these proteins has been shown by using pharmacological inhibitors in in vitro scratch assays, together with expression protein level and phosphorylation studies by western blot [20]. First of all, the use of an EGF receptor inhibitor (EGFRi, PD153035) has been reported to be a strong inhibitor of epithelial cell migration [20,56,67]. Indeed, the high migratory percentage with OA/HP-β-CD complexes observed in the scratch was also significantly abrogated by EGFRi (Figure 2a,b).

In contrast, the use of specific inhibitors against MEK and JNK partially blocked OA/HP-β-CD as well as OA/DMSO migratory activity. Thus, in any case, OA/HP-β-CD complexes migration activity was significantly compromised, in a very similar fashion observed with OA/DMSO with the different inhibitors. All these data suggest that OA/HP-β-CD complexes seemed to use triggered OA-enhanced biochemical pathways as OA/DMSO.

### 3.6. OA/HP-β-CD Complexes Induce c-Jun Phosphorylation Transcription Factor at Cells at Scratch Edge in Mv1Lu

The stimulation of OA produces an activation of c-Jun that is observed at the nuclear level. Moreover, migrating cells show an activation of transcription factor c-Jun at the nuclei in response to OA stimulation [21]. We monitored p-c-Jun subcellular localization along the scratch edge. Strikingly, cells at scratch edge overexpressed p-c-Jun at cell nuclei in response to OA/HP-β-CD complexes, similarly to the OA/DMSO condition (Figure 3).

Interestingly enough, OA/HP-β-CD was causing a wider recruitment of migrating cells in the scratched area compared to OA/DMSO. Although the quantification of c-Jun phosphorylation showed that OA/DMSO treatment produced a strong stimulation of p-c-Jun at the edge, OA/HP-β-CD treatment produced a higher intensity of p-c-Jun at nuclei, together with a greater number of cells revealing p-c-Jun (Appendix A); the calculated ratio of positive p-c-Jun nuclei versus the total number of nuclei in the picture also showed a higher ratio of the OA/HP-β-CD compared to the OA/DMSO (Appendix A). Additionally, we noticed that nuclear p-c-Jun intensity was higher at the scratch edge cells and decreased as the cells were far from the scratch edge, exhibiting a p-c-Jun intensity gradient, positively correlating with cells with a high migratory status.

### 3.7. OA/HP-β-CD Complexes Promote Changes in Actin Fibers and Paxillin Distribution in Mv1Lu Cells

We have recently observed that OA effects on cell migration include cytoskeleton and focal adhesion (FA) remodeling [21]. To assess the consequences of complexed OA with HP-β-CDs, we carried out immunocytochemistry assays in in vitro scratched Mv1Lu cells targeting actin fibbers (F-Actin) and paxillin. Just after scratching (time 0 h), cells displayed paxillin as compact FA (Figure 4), and they exhibited low FA density and size.

Strikingly, after 6 h with OA/HP-β-CD treatment, we observed a high FAs density that was very noticeable at the pictures shown, being also coherent with the effects seen with OA/DMSO, however, with a slightly higher tendency than with OA/DMSO. Moreover, in terms of FAs size at 6 h, no statistical differences were observed between basal control and both OA/DMSO and OA/HP-β-CD. Furthermore, after 12 h, FAs size was significantly smaller than basal control in OA/DMSO and OA/HP-β-CD conditions. Remarkably, we noticed that in the OA/HP-β-CD condition after 6 and 12 h exhibited a higher cell spreading along the scratch edge than with OA/DMSO when we took a look with more magnification (Appendix A). Interestingly, neither FAs size nor FAs density showed statistically significant differences between OA/DMSO and OA/HP-β-CD conditions. Regarding actin fibers, noticeable changes were observed in all conditions after 6 and 12 h; however, they were very evident after 12 h with OA/DMSO and OA/HP-β-CD, a time when cells displayed a lower actin intensity due to the actin disassembling during cell movement (Appendix A) [68].

All these observations suggest that similarly to OA/DMSO, OA/HP-β-CD complexes produce high dynamization of the cytoskeleton and FA, compatible with a neat increment in migration rate.

### 3.8. OA/HP-β-CD Complexes Stimulate Migration in MDA-MB-231 Cells while Minimizing OA/DMSO Cytotoxic Effects

It has been seen that OA/DMSO stimulates MDA-MB-231 [23,35,69] migration in in vitro scratch assays at an optimal concentration of 10 µM [20,21]. We tested several concentrations of OA/HP-β-CD complexes, where MDA-MB-231 cells showed significant cell migration compared to basal (C) and vehicle control HP-β-CD conditions (Figure 5a,b).

Similar migration percentage values were observed between OA/DMSO and all OA/HP-β-CD concentrations tested, with no statistically significant differences reached. Nevertheless, there was a higher migration activity tendency within 17/4.25 µM/mM and 21/5.25 µM/mM OA/HP-β-CD concentration complexes. Furthermore, it was easy to notice that, in the case of OA/HP-β-CD, more cells could be found reaching the scratched gap area than in the OA/DMSO condition (Figure 5a).

OA/DMSO exhibits a mild antiproliferative effect on MDA-MB-231 cells [20]. OA/HP-β-CD treatment showed a nearly identical cell count to basal (control) and vehicle controls treatments in contrast to OA/DMSO, which was slightly affected since a lower number of cells was observed, a trend that was held for 10 days. Interestingly, these data suggest that MDA-MB-231 cell proliferation performed better, and near control samples, for OA/HP-β-CD complexes than for free OA/DMSO samples (Figure 5c).

All together, these experiments suggest that complexed OA with HP-β-CDs enhanced MDA-MB-231 cell migration while improving cell viability when compared to OA/DMSO.

### 3.9. Signaling Pathways Regulated by EGFR and c-Jun Activation, Necessary for Cell Migration, Are Induced by OA/HP-β-CD Complexes in MDA-MB-231 Cells

MDA-MB-231 cells are a suitable epithelial model for epidermal growth factor receptor (EGFR) expression regulation studies [21,23,70]. EGFR inhibitor (EGFRi), PD153035, is capable of reducing migration in MDA-MB-231 cells in OA/DMSO [20,21] and also in OA/HP-β-CD stimulated cells (see Figure 2).

With the objective of deeply deciphering the molecular mechanisms behind the OA/HP-β-CD complexes effect on cell migration, we decided to set up a whole study of key regulatory proteins involved in cell migration by using sub confluent cells [19,21,71]. Cells were stimulated with 10 µM OA/DMSO and 21/5.25 µM/mM OA/HP-β-CD, and the activation of different proteins was evaluated. Mainly, the phosphorylation consequences of all proteins studied were milder when cells were stimulated with OA/HP-β-CD in comparison to their DMSO-solubilized counterparts. To begin with, a p-EGFR stimulation was detected 2 h after treatment with OA/HP-β-CD, later increasing at 3 and 4 h; however, when compared to OA/DMSO, the stimulation was softer (Figure 6a,b).

When looking at a downstream EGFR kinase, p-ERK1/2, it was increased by OA/HP-β-CD treatment after 2 h following the same trend as EGFR phosphorylation (Figure 6a,b). Interestingly, OA/HP-β-CD showed significantly lower intensity values at time 2 h, and they were kept higher for longer periods of time. Regarding p-JNK1/2, both OA/HP-β-CD and OA/DMSO-induced stimulation stimulated this kinase at time 1 h; however, later in time, the dynamics were dissimilar. Interestingly, when blotting total c-Jun and active phosphorylated (p)-c-Jun forms, we observed changes on both antigens in response to OA/HP-β-CD, an effect that is evident also in the case of OA/DMSO [21]. In particular, in the case of p-c-Jun, the treatment with OA/HP-β-CD showed a mild increase at time 2 h that was sustained up to 6 h. In contrast, OA/DMSO produced a stronger stimulation of p-c-Jun at time 1 h; it was sustained in time up to 6 h and exhibited higher values than cyclodextrin-complexed OA. In any case, no increases in p-JNK1/2 and p-c-Jun regulatory proteins were observed with DMSO and HP-β-CD vehicle controls. Finally, total c-Jun levels were clearly increased by OA/HP-β-CD with a 2 h delay (3, 4, and 6 h) when compared to OA/DMSO, which stimulated c-Jun overexpression at the early time of 1 h. The overexpression and activation of the above-mentioned proteins in response to OA/HP-β-CD imply their participation in the cell migration phenomenon promoted by cyclodextrin-complexed OA.

OA/DMSO stimulation of Mv1Lu shows similar molecular mechanisms as the MDA-MB-231 cell line [21]. Thus, we studied changes at the level of protein profiling in response to OA/HP-β-CD and compared them to OA/DMSO (Appendix A). Once again, OA/HP-β-CD complexes were capable of stimulating EGFR with similar phosphorylation patterns between OA/DMSO and OA/HP-β-CD and only differing in intensity values (Appendix A). Strikingly, when looking at p-ERK1/2, OA/HP-β-CD stimulated p-ERK1/2 in a manner that differed from OA/DMSO, since p-ERK1/2 levels were significantly increased at earlier times (1 and 2 h) while OA/HP-β-CD produced a later response at times 4 and 6 h (Appendix A). OA/DMSO and OA/HP-β-CD showed the same phosphorylation dynamic for p-JNK1/2, noticing in both cases an early response (time 1 h) with the highest levels of p-JNK1/2. Finally, c-Jun activation and overexpression were enhanced by OA/HP-β-CD complexes. Regarding p-c-Jun, OA/HP-β-CD showed a clear increase on this active form such as OA/DMSO, but, remarkably, this increase was sustained until later times. In addition to this, c-Jun total levels were slightly increased by OA/HP-β-CD at time 1 h and then at time 4 and 6 h, remaining as a trend compared to basal conditions, as this induction on total c-Jun was not as higher as with OA/DMSO.

Altogether, these results on protein expression and activation by phosphorylation on MDA-MB-231 and Mv1Lu cells point to the fact that complexed OA with modified cyclodextrins, as seen with OA/DMSO, clearly triggered different protein kinase targets to induce biochemical pathways, which are necessary to lead to cell migration.

## 4. Discussion

Oleanolic acid-triggered molecular mechanisms are compatible with improved cell migration, making them a potential wound healing agent [20,21]. Nevertheless, its hydrophobic nature stunts its application in hydrophilic contexts [52]. In this paper, we have shown how OA complexation in HP-β- and HP-γ- modified cyclodextrins can improve OA application and OA activity on in vitro cell culture models Mv1Lu and MDA-MB-231 [20,21].

The lipophilic nature of OA implies the need for improved delivery with new carriers, which could allow its conservation, protection, and application. For OA delivery, many studies have shown its vehiculation by a variety of macromolecules in several applications. For instance, nanoparticles made from polymers such as polyethylene glycol, cellulose, or silicone have been used to encapsulate OA [72,73,74,75,76]. Despite this, these nanoparticles require a long chemical synthesis and further 3D structure analyses, which are expensive and time-consuming. That is why cyclodextrins might be considered a better option to improve OA delivery and application. Remarkably, a dehydration step, performed by freeze drying, improves the effect on OA from cyclodextrins since they protect and preserve OA in long-term storage by keeping OA loaded for weeks at 4 °C and even at 20 °C. Conversely, OA/DMSO required specific storage conditions in liquid nitrogen (−190 °C) to maintain it long-term [20,21]. What is more, the OA/CDs powder formulation can be easily diluted in cell culture media or other desired aqueous solutions, avoiding microbial contamination and preserving the osmolarity of cell culture media.

HP-β- and HP-γ-CDs were chosen for OA complexation and subsequent in vitro assays for their improved water solubility compared to native cyclodextrins [48,77]. Both HP-β- and HP-γ-CD have shown that they can properly encapsulate OA, and dried solid complexes of OA can be obtained. However, it has been noticed that there is different behavior between them. The higher encapsulation constant (K_c_) of the HP-γ-CD (K_c_ = 645 M^−1^) complexes compared to the HP-β-CD complexes (K_c_ = 201 M^−1^) [52] is associated with higher complex stability. This could contribute to a reduced loss of OA during the encapsulation and dehydration processes of the complex, resulting in higher encapsulation efficiency (EE) and a higher final retention of the active compound. As a consequence, this leads to a higher drug loading (DL mg/g) in the ultimately obtained solid complexes of OA/HP-γ-CD. This may represent an advantage at first, but it is necessary to determine whether the stability of the complex, dependent on its K_c_ value, allows a proper release of the host compound into the medium so that it can exert its biological activity.

The stability of the obtained complexes, defined by the K_c_ values of the complexes and dependent on the type of CD, can modulate the activity of the host molecule of the complex. In this regard, the results of cell migration show that, for the same concentration of 12.5 μM of OA, the level of cell migration has been higher for HP-β- than for HP-γ-CD complexes. The complex OA/HP-β-CD has a lower complexation constant (K_c_) [52], indicating that OA is released more easily from the hydrophobic cavity of the CD, as it is a less stable complex compared to the one formed with HP-γ-CD. For this reason, HP-β-CD appears to be the most suitable CD for the complexation of OA, as it demonstrates a better compromise between adequate compound release capability and the high aqueous solubility of OA.

Furthermore, the results obtained demonstrate that the complexation of OA in CD partially limits its ability to stimulate cell migration, as it is necessary to increase the concentration of OA to 12.5 μM and 62.5 μM to achieve the same level of cell migration obtained with 5 μM of OA dissolved in DMSO. This limitation in the biological activity of different encapsulated active compounds, previously described by other authors [43,78,79,80], is expected since encapsulation traps the active molecule and its release is conditioned by the stability of the complexes. On the other hand, it is known that, when solubilized with DMSO, OA acts at a very tight range of concentrations to see its cell migration activity on Mv1Lu and MDA-MB-231 epithelial cell lines [20,21]. Concentrations out of this range have cytotoxic effects and cell viability loss. Indeed, OA biological effects are cell-type specific, as OA is used as an antiproliferative in some tumor cell lines [62,81,82]. With our gathered data on scratch and proliferation assays, we observed that cyclodextrin complexation changed the OA dose-effect. First, OA/HP-β-CD complex concentrations that show cell migration activity peaks were mainly two times higher than the ones with OA/DMSO for both cell lines Mv1Lu and MDA-MB-231. Additionally, proliferation assays in these two cell lines showed a slightly higher cell count than OA/DMSO conditions, suggesting a protective effect of the cyclodextrin. These data could be explained by the non-direct and slow OA delivery to cells because of the OA lodging into the cyclodextrins, which produces a sustained OA release [83]. This improved delivery suggests better OA bioavailability for epithelial cells, which in fact could mitigate cell viability loss and improve cell migration from the scratch edges.

We noticed that complexed OA showed reinforced cell migration activity on Mv1Lu. On in vitro scratch assays, we observed that scratch treated with OA/HP-β-CD complexes displayed higher recruitment of migrating cells than OA/DMSO, as a migration morphology was very patent in several lines from the scratch edge. Even though this recruitment was less patent, it could also be seen in MDA-MB-231 scratch assays. Interestingly, this phenomenon correlated positively with immunocytochemistry assays in Mv1Lu, in which we study p-c-Jun activation and its subcellular location, which are crucial to express cell migration genes [84,85]. We saw a different pattern of activation when comparing OA/DMSO and OA/HP-β-CD scratched cells. Strikingly, scratched cells treated with OA/HP-β-CD showed a wider belt of cells with nuclear overexpressed and activated c-Jun. Indeed, the ratio of positive nuclei and total nuclei was higher in the OA/HP-β-CD condition than in the OA/DMSO condition, showing a decrease as the cells became far from the scratch edge. These results point out that OA complexation with cyclodextrins endows the oleanolic with improved properties on cell migration.

In terms of paxillin remodeling, OA/DMSO and OA/HP-β-CD conditions displayed quite similar patterns, especially when looking at focal adhesion size. We could not identify differences between the two treatments, so more studies must be carried out to decipher these mechanisms, such as paxillin/FAK colocalization assays by confocal microscopy [21]. However, regarding F-Actin distribution at cells at scratch edges, we could observe filopodia and ruffle formation during OA-triggered migration. Indeed, Mv1Lu scratch-edge cells treated with OA/HP-β-CD showed qualitatively a greater number of filopodia and ruffles since the cells were more spread along the scratched area. This, together with the fact that in OA/HP-β-CD condition F-Actin filaments were less patent than in OA/DMSO condition, suggests a highest degree of mobilization of structures in response to oleanolic when it was complexed. This could be related to the activation of c-Jun, the master regulator of cell migration [26,84,85], which lasted longer when the cells were stimulated with OA/HP-β-CD.

Some of the molecular events happening in cells when they are stimulated with OA suggest a general activation of several kinases related to cell movement. In order to deepen the molecular mechanisms underlaying the effects of OA/HP-β-CD complexes on cell migration, the same key regulatory proteins were examined [23,84,85,86,87,88,89]. Generally speaking, the responses obtained with OA/HP-β-CD were similar to those obtained with OA/DMSO. Surprisingly, the activation (phosphorylation) of many of the different proteins studied was milder with complexed OA compared to its DMSO-solubilized counterpart. However, in the case of MDA-MB-231, the activation pattern showed a delayed response in the OA/HP-β-CD condition than in the OA/DMSO condition. Furthermore, the levels of the proteins assayed were lower in the OA/HP-β-CD condition than in the OA/DMSO condition, probably due to the different dynamics that cyclodextrins brought.

What is more, this study found that cell migration promoted by OA/HP-β-CD complexes in Mv1Lu cells exhibited similar molecular mechanisms to MDA-MB-231 cells, as it was previously described for OA/DMSO [21]. Interestingly, as seen with OA/DMSO, the cell migration that OA/HP-β-CD complexes produced was compromised when specific inhibitors against EGFR, MEK, and JNK were added to the media in scratch assays. Moreover, similar phosphorylation patterns were observed between OA/DMSO and OA/HP-β-CD for p-EGFR and p-JNK proteins. By contrast, OA/HP-β-CD complexes stimulated p-ERK1/2 at later times compared to OA/DMSO. Regarding c-Jun and p-c-Jun blots, the expression of both forms was increased and sustained at later times in the OA/HP-β-CD condition. These results suggest a sustained effect of complexed OA on this transcription factor, which also correlates positively with immunocytochemistry assays, where a greater number of cells with high p-c-Jun were observed at scratch edges. Altogether, these observations in western blots led us to think that these regulator proteins show a delayed response with OA complexes, probably due to the cyclodextrin slow-release dynamic. This response could be beneficial because, over time, OA complexes provided more migrating cells than OA/DMSO covering the scratched area.

Altogether, the evidence shown by the in vitro scratch assays favors the higher efficacy of cyclodextrins delivering OA to cells, based on the wide spread of activation caused. However, to make a perfect correlation, a western blot should be performed on the area that is activated by OA/HP-β-CD and compared to the rest of the epithelia, but, at present, we have not found a way of analyzing the samples in such a way. This may reflect the fact that using confluent cultures to depict what is happening at the scratch edge has some limitations that, with the current technology, cannot been overcome.

In summary, the study on key regulatory proteins indicates that complexed OA with modified cyclodextrins triggers the same biochemical pathways as OA/DMSO, which is necessary for cell migration of both MDA-MB-231 and Mv1Lu cells. In this study, we highlighted the potential of these epithelial cell models for investigating EGFR and key regulatory kinases expression in cell migration under OA/HP-β-CD complexes effects.

All in all, here we showed how cyclodextrin complexation allows OA application improvement, making it suitable to develop a product that could be used topically on skin for wound healing. Given the enhanced OA solubilization and improved activity on epithelial cells, we are prone to performing in vivo assays in the future. In this sense, many assays can be carried out on in vivo models to study complex OA effects on wound healing [90,91].

In this paper, we have deciphered how OA complexation with modified cyclodextrins could improve OA application and activity in vitro. Our results showed that OA/CDs, mainly OA/HP-β-CD complexes, showed significant activity on cell migration and toxicity protection in Mv1Lu and MDA-MB-231 cells. In addition to this, OA/HP-β-CD complexes induce a whole signaling system compatible with cell migration, such as OA/DMSO. Even though both agents triggered the same mechanisms, OA/HP-β-CD complexes showed greater migrating cell recruitment powered by greater c-Jun activation and displaying a sustained OA effect on cell migration provided by cyclodextrins slow release to the media. Validating that OA maintains its biological activity when it is complexed in the form of a solid complex with HP-β-CD is a key aspect of drug application because solid-state complexes improve the handling and stability of compounds, enabling better dose standardization [43]. This represents a clear technical advantage for the pharmaceutical industry. Furthermore, the complexation of OA with CD enables achieving effective OA concentrations in an aqueous medium, eliminating the need for organic solvents such as DMSO. Finally, given the implications that this paper shows, the research on OA complexed with cyclodextrins for developing an effective treatment for difficult wounds seems very promising.

## 5. Conclusions

Oleanolic acid conjugated to cyclodextrins (OA/CDs) exhibits enhanced properties compared to OA solubilized with DMSO. Specifically, OA/HP-β-CD not only enhances cell migration but also improves cell viability. It also plays a role in cytoskeleton reorganization and in mobilizing focal adhesions. Much like OA/DMSO, OA/HP-β-CD activates specific migration pathways. Moreover, complexing OA with cyclodextrins significantly enhances the molecule’s stability and enables its solubility in aqueous solutions. This crucial enhancement extends its practical applications. Consequently, we now have a hydrophilic conjugate that is more bioavailable, stable, and readily applicable to wound care.

## Figures and Tables

**Figure 1 ijms-24-14860-f001:**
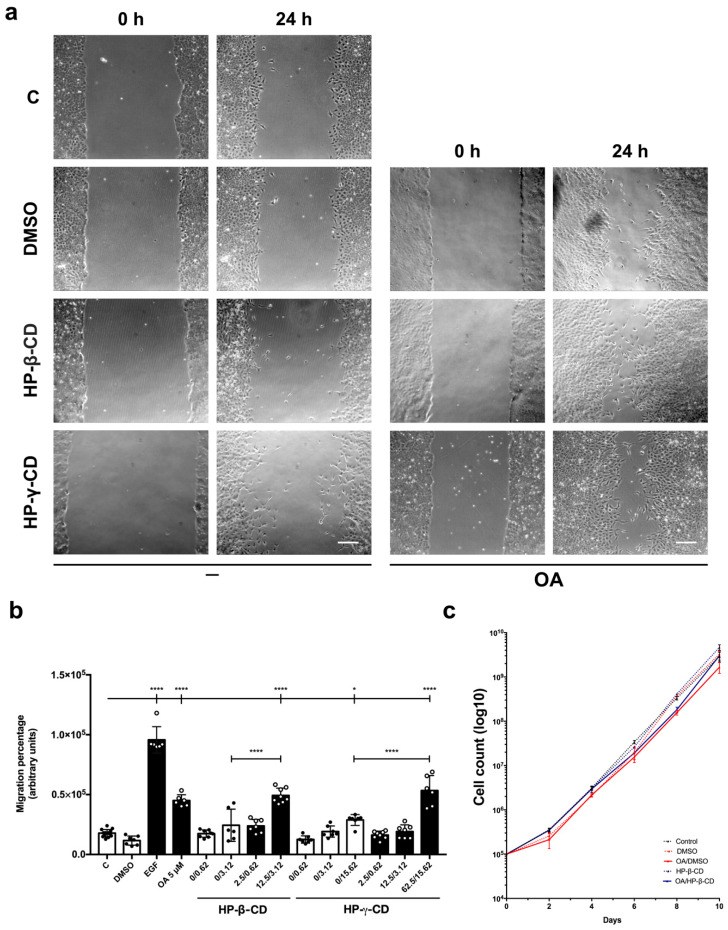
Freeze-dried OA/HP-β-CD and OA/HP-γ-CD complexes stimulate migration in Mv1Lu cells, minimizing OA/DMSO cytotoxic effects. Confluent Mv1Lu cells were scratched with a pipette tip and allowed to migrate for 24 h. (**a**) Representative images of the in vitro scratch assay with cell migration under basal conditions (C) compared to those with 5 µM OA/DMSO, 12.5/3.12 µM/mM OA/HP-β-CD, and 62.5/15.62 µM/mM OA/HP-γ-CD; after 24 h treatment. Equivalent concentrations of DMSO, HP-β-CD, and HP-γ-CD were used as vehicle controls. The scale bar indicates 200 µm. (**b**) Plot represents cell migration as the difference between areas at time 0 h and time 24 h in each condition, named as migration percentage. Control conditions: white bars, black dots. Stimuli: black bars, white dots. The X axis indicates treatment conditions; for those with HP-β-CD and HP-γ-CD, the concentrations are represented as the molar ratio OA µM/CD mM. Epidermal growth factor (EGF) was added at 10 ng/mL as a positive migration control. Asterisks indicate statistically significant differences between conditions according to a one-way ANOVA statistical analysis (* *p* < 0.05 and **** *p* < 0.0001). (**c**) The long-term effects of 5 µM OA/DMSO and 12.5/3.12 µM/mM OA/HP-β-CD (with vehicle controls) on Mv1Lu proliferation were assessed by counting total cells at the indicated times. The logarithm of the mean number of cells against time is plotted.

**Figure 2 ijms-24-14860-f002:**
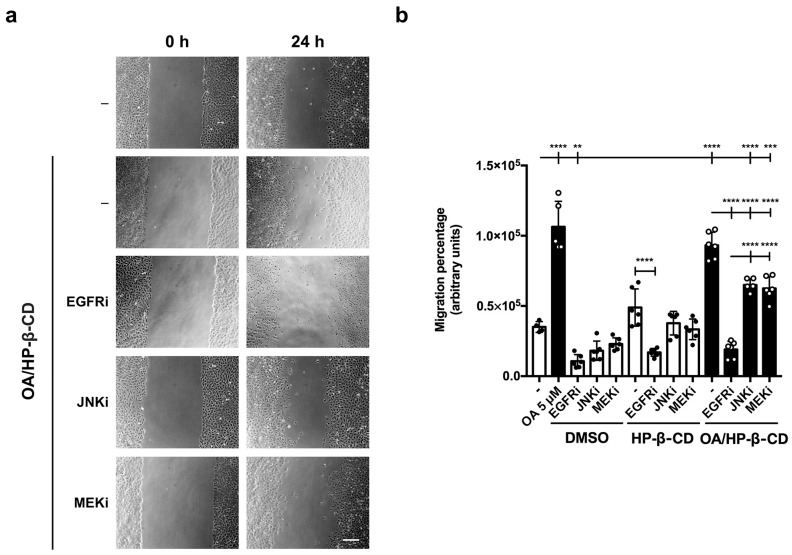
Pharmacological inhibitors against migration-related proteins decrease OA/HP-β-CD-induced cell migration in scratched Mv1Lu cells. Confluent Mv1Lu cells were scratched with a pipette tip and allowed to migrate for 24 h. Before adding treatments, EGFR, JNK, and MEK inhibitors were added to the medium (EGFRi, epidermal growth factor receptor inhibitor; JNKi, c-Jun N-terminal kinase inhibitor; MEKi, mitogen-activated protein kinase inhibitor). (**a**) The figure shows representative images of the in vitro scratch assay with cell migration under basal conditions (C) compared to those with 12.5/3.12 µM/mM OA/HP-β-CD alone (-) or with the above-mentioned inhibitors after 24 h treatment. The scale bar indicates 200 µm. (**b**) Plot represents cell migration as the difference between areas at time 0 h and time 24 h in each condition, named as migration percentage. Control conditions: white bars, black dots. Stimuli: black bars, white dots. Equivalent inhibitor concentrations were added too in vehicle control conditions DMSO and HP-β-CD. Asterisks indicate statistically significant differences between conditions according to a one-way ANOVA statistical analysis (** *p* < 0.005, *** *p* < 0.001, and **** *p* < 0.0001).

**Figure 3 ijms-24-14860-f003:**
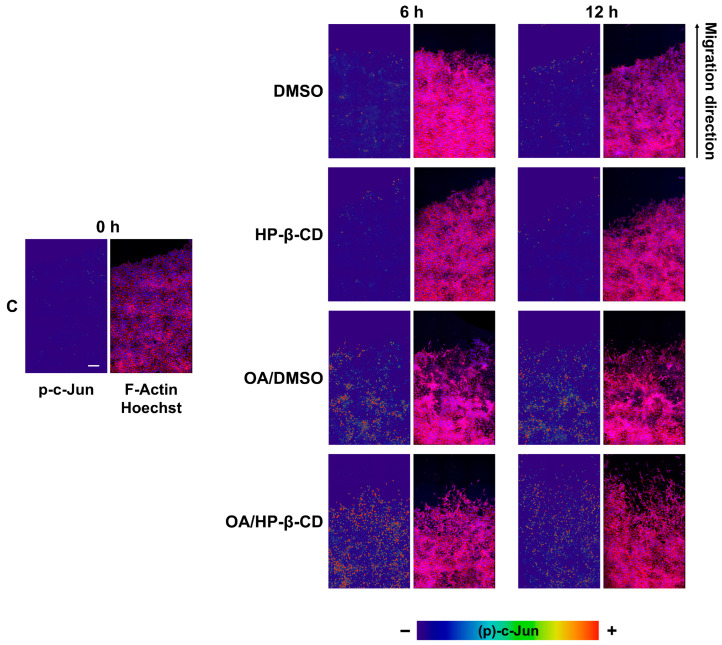
OA/HP-β-CD complexes promote c-Jun phosphorylation at the edge of scratched Mv1Lu cells. Confluent Mv1Lu cells were scratched and allowed to migrate for the indicated times (6 and 12 h). Cells were treated with 5 µM OA/DMSO or 12.5/3.12 µM/mM OA/HP-β-CD. Equivalent concentrations of DMSO and HP-β-CD were used as vehicle controls. Cells were immunostained with specific antibodies against c-Jun active phosphorylated form (p-c-Jun). Co-staining with phalloidin and Hoechst-33258 was used to show the actin cytoskeleton and nuclei, respectively. Images of p-c-Jun fluorescence were converted into pseudo-color with ImageJ software to show the intensity of p-c-Jun staining. Actin fibers (F-actin): red. Nuclei: blue. Images were obtained with a confocal microscope. This experiment was repeated at least three times. Representative images are shown. Scale bar indicates 100 µm.

**Figure 4 ijms-24-14860-f004:**
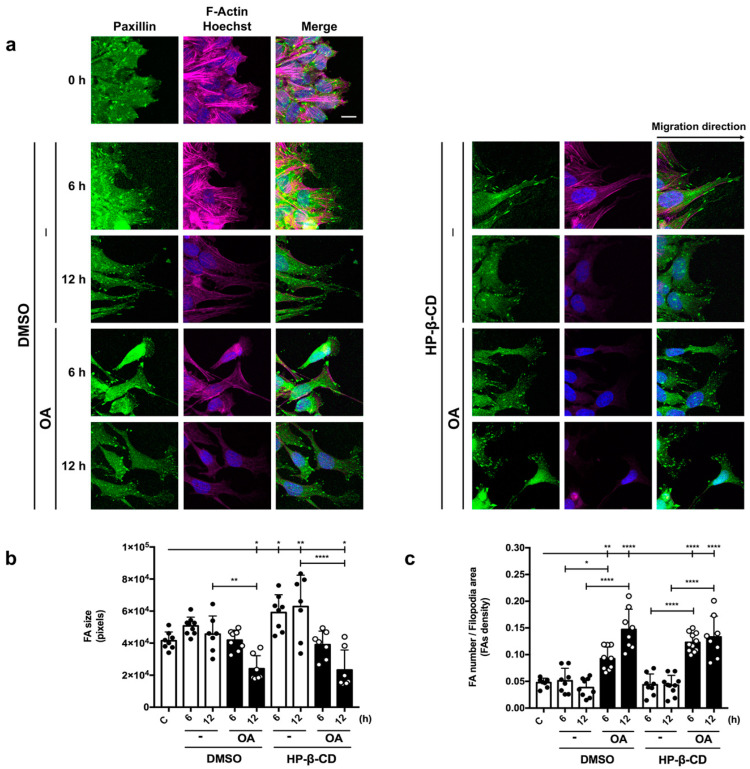
OA/HP-β-CD complexes promote changes in focal adhesions (FAs) revealed by Paxillin. (**a**) Confluent Mv1Lu cells were scratched and allowed to migrate for 6 and 12 h. Cells were treated with 5 µM OA/DMSO or 12.5/3.12 µM/mM OA/HP-β-CD. Equivalent concentrations of DMSO and HP-β-CD were used as vehicle controls. Cells were immunostained with specific antibodies against paxillin. Co-staining with phalloidin and Hoechst-33258 was used to show the actin cytoskeleton and nuclei, respectively. Paxillin: green. Actin fibers (F-Actin): red. Nuclei: blue. Images were obtained with a confocal microscope. Insets corresponding to 1/9 of the 63× original images (Appendix A) are shown for a detailed view of paxillin structures. This experiment was repeated at least three times. The scale bar indicates 6 µm. (**b**) FA size (average size) at the filopodia area. (**c**) Quantification of the density of FA (as FA number per filopodia area). Control conditions: white bars, black dots. Stimuli: black bars, white dots. A one-way ANOVA statistical analysis was performed (* *p* < 0.05, ** *p* < 0.005 and **** *p* < 0.0001).

**Figure 5 ijms-24-14860-f005:**
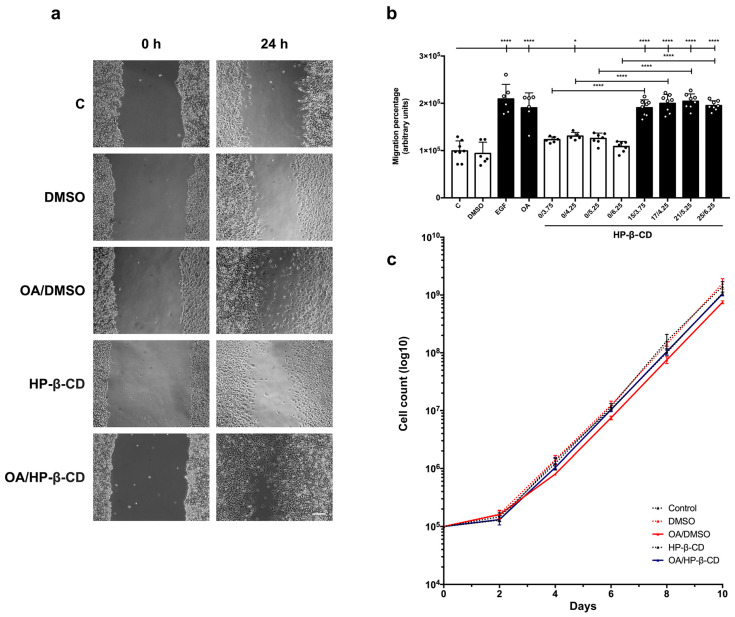
OA complexes with modified cyclodextrins HP-β-CD promote cell migration in MDA-MB-231 cells. Confluent MDA-MB-231 cells were scratched with a pipette tip and allowed to migrate for 24 h. (**a**) Representative images of the in vitro scratch assay with cell migration under basal conditions (C) compared to those with 5 µM OA/DMSO, 21/5.25 µM/mM OA/HP-β-CD, and equivalent concentrations of DMSO or HP-β-CD after 24 h treatment. The scale bar indicates 200 µm. (**b**) Plot represents cell migration as the difference between areas at time 0 h and time 24 h in each condition, named as migration percentage. Control conditions: white bars, black dots. Stimuli: black bars, white dots. The X axis indicates treatment conditions; for those with HP-β-CD concentrations are represented as the molar ratio OA µM/CD mM. Epidermal growth factor (EGF) was added at 10 ng/mL as a positive migration control. Asterisks indicate statistically significant differences between conditions according to a one-way ANOVA statistical analysis (* *p* < 0.05 and **** *p* < 0.0001). (**c**) The long term effects of 5 µM OA/DMSO and 21/5.25 µM/mM OA/HP-β-CD (with vehicle controls) on MDA-MB-231 proliferation were assessed by counting total cells at the indicated times. The logarithm of the mean number of cells against time is plotted.

**Figure 6 ijms-24-14860-f006:**
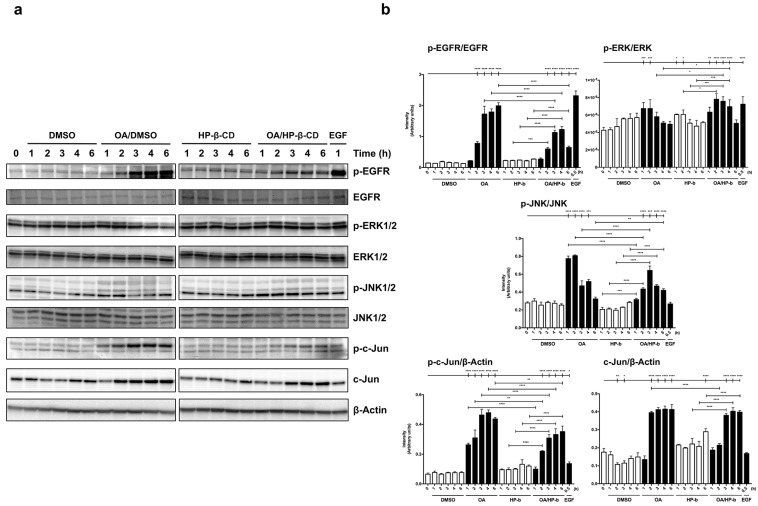
Signaling pathways regulated by EGFR and c-Jun activation, necessary for cell migration, are induced by OA/HP-β-CD complexes in MDA-MB-231 cells. (**a**) Total protein extracts from serum-starved sub-confluent MDA-MB-231 cells treated with 10 µM OA/DMSO, 21/5.25 µM/mM OA/HP-β-CD, or 10 ng/mL EGF. These extracts were assayed at the indicated times (hours) targeting: phospho-EGFR (Tyr 1068), phospho-ERK1/2 (Thr 202/Tyr 204), phospho-JNK1/2 (Thr 183/Tyr 185) and phospho-c-Jun (Ser 63). Total protein expression was assayed for the above-mentioned active protein forms: EGFR, ERK1/2, JNK1/2, and c-Jun. β-Actin was used as a loading control. DMSO and HP-β-CD equivalent concentrations were added as vehicle controls. A representative experiment was shown. (EGFR, epidermal growth factor receptor; ERK1/2, extracellular signal-regulated kinases 1 and 2; JNK1/2, c-Jun N-terminal kinases 1 and 2). (**b**) Column bar graphs represent the intensity values of each protein assayed by western blot by collecting the data from three independent experiments. Intensity values were quantified and gathered by ImageJ software. Asterisks indicate statistically significant differences between the selected conditions according to a one-way ANOVA statistical analysis. (**p* < 0.05, ***p* < 0.005, ****p* < 0.001, and *****p* < 0.0001).

**Table 1 ijms-24-14860-t001:** Hydroxypropyl beta and gamma cyclodextrins (HP-β- and HP-γ-CDs) show high rates od complexation parameters with oleanolic acid (OA). Dehydration yield (DY), encapsulation efficiency (EE), and drug load (DL) of solid complexes obtained by freeze drying for OA encapsulated with HP-β- or HP-γ-CDs. Values represent means of triplicate determination; the data presented in the table represent the mean ± standard error of the mean.

	DY (%)	EE (5%)	DL (mg/g)
**HP-** **β-CD**	89.4 ± 6.7	80.0 ± 7.6	1.06 ± 0.07
**HP-** **γ-CD**	91.6 ± 0.7	91.9 ± 9.2	1.17 ± 0.18

**Table 2 ijms-24-14860-t002:** OA at freeze-dried OA/HP-β-CD complexes is stable at room temperature. The table shows OA analysis by HPLC to determine the OA amount in OA/HP-β-CD complexes, kept at different temperatures. Samples from these complexes were taken at 1, 2, 4, and 8 weeks after encapsulation. The table shows the OA areas corresponding to the absorbance peaks detected at 210 nm in the HPLC system. On the other side, the table indicates the amount of sample taken from the solid freeze-dried OA/HP-β-CD complexes. OA quantification was expressed as OA mg per HP-β-CD g, defined as drug load (DL). The data presented in the last row (Global) represents the mean ± standard error of the means of the different times (0 to 8 weeks).

Weeks after Encapsulation (w)	Storage Temperature (°C)	OA Peak Area at A_210nm_	Solid Complex Weight (mg)	DL (mg/g)
**0 w**	20	489.0	21.3	1.11
	20	468.0	20.1	1.13
**1 w**	20	415.2	19.0	1.06
	4	517.6	21.0	1.19
	−80	527.5	22.0	1.16
**2 w**	20	489.8	21.4	1.11
	4	467.4	21.7	1.04
	−80	426.3	20.2	1.02
**4 w**	20	505.0	21.9	1.12
	4	501.0	22.2	1.09
	−80	514.0	21.9	1.14
**8 w**	20	459.6	20.2	1.10
	4	431.2	19.6	1.07
	−80	447.0	20.3	1.07
**Global**	20	471.25 ± 29.10	20.65 ± 0.98	1.11 ± 0.02
	4	479.30 ± 3.14	21.13 ± 0.98	1.10 ± 0.06
	−80	478.70 ± 42.95	21.10 ± 0.85	1.10 ± 0.06

## Data Availability

All data supporting this paper had been included. In order to clarify western blot data, we displayed cropped blots of each figure in the paper as (Appendix A). These figures contain original blots of each protein assayed with molecular weight markers for a better identification.

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
