# Peer review of "Oleanolic Acid Complexation with Cyclodextrins Improves Its Cell Bio-Availability and Biological Activities for Cell Migration"

_ijms, 2023, doi:10.3390/ijms241914860_

Round 1

Reviewer 1 Report

The authors have provided a comprehensive biological insight in their paper titled "Oleanolic Acid Complexation with Cyclodextrins Improves its Cell Bioavailability and Biological Activities for Cell Migration." The study investigates the impact of oleanolic acid (OA) formulation with cyclodextrin on various biological aspects. While the paper is commendable, there are a few recommendations and concerns to consider.

In the abstract, it is suggested that the introductory section from lines 10 to 18 be condensed, highlighting the key findings of the study.

In the introduction (lines 67-93), the authors should provide a concise overview of the studies conducted and briefly outline the information gained from these investigations.

Regarding Figure 1, the images are small and challenging to interpret. It is recommended to enlarge Figure 1A and place Figures 1B and 1C beneath Figure 1A, sharing half the page.

On page 8, line 330, the statement, "12.5/3.12 µM/mM and 125/31.25 µM/mM for OA/HP-CD and OA/HP-CD respectively," should be simplified, possibly using v/v, w/w, or mole ratio for clarity.

The paper should include DLS profiles of the formulation to ensure formation and stability testing.

The authors should add a separate conclusion section to summarize their findings effectively.

On page 3, line 104, it is unclear why OA and CD are mixed in a molar ratio of 0.2/50 if they form a 1:1 complex. Line 105 should be elaborated upon to clarify the concept of phase solubility, and the reasons for selecting the values 0.2/0.5 should be explained.

Considering the variability in chemical purity, instrument parameters, and solution handling, it is recommended that in section 2.2, after mixing the drug and solubilizing agent, every formulation should be verified using DLS size and zeta potential measurements before and after encapsulation to confirm drug loading. The formulation mixture should then be purified through a dialysis membrane. Depending on the drug molecule, a calibration curve could be constructed using UV or HPLC methods. This step is essential to validate the study.

Lastly, the authors should provide their rationale for choosing the freeze-drying method, as it pertains to the study's objectives and the properties of the formulations.

Reviewer 2 Report

The manuscript entitled "Oleanolic acid complexation with cyclodextrins improves its cell bio-availability and biological activities for cell migration" compares oleanolic acid (OA) dissolved in DMSO with OA-cyclodextrin complexes in terms of their effects on cell migration using mink lung epithelial cells and human mammary adenocarcinoma cells. The study is implemented very well, and the results are interpreted in a reasonable manner. However, although the authors in the Introduction, as well as, in the Abstract and Discussion (especially lines 706-708) imply that they aim in using OA in wound healing interventions (e.g. against chronic ulcers), they have used cell types irrelevant with wound healing. I would suggest performing at least some basic experiments, such as a scratch assay in human normal keratinocytes or at least in immortalized HaCaT cells.

It is also important to note, that the scratch assay has nothing to do with wound healing, it is simply a measure of cell migration. Hence, the authors should avoid using this term, e.g. the subtitle "2.7. Wound healing scratch assay" should be corrected to "2.7. In vitro scratch assay" and the corresponding changes should also be applied in lines 54, 314, 383, 395, 440, 475, 483, 639, 693, etc.

Another weakness of the manuscript is that the Abstract is very general, devoting 14 lines on the background and only 4 lines on the actual findings and conclusions of the present study.

There are a only few minor errors, such as:

line 67, instead of "lyophilic" should read "lipophilic"

line 586, instead of "polietilenglicol" should read "polyethyleneglycol"

line 626, instead of "has" should read "have"

line 641, the word "like" should be deleted

Round 2

Reviewer 1 Report

The authors have considered the suggestions and made the required modifications. The paper could be accepted in its present form.

Reviewer 2 Report

The manuscript is now suitable for publication.